# Tracking In-Situ Snow Accumulation at Neumayer, Coastal Antarctica: Signs of Climatic Changes in the past 30 Years?

Valerie Reppert<sup>1</sup>, Olaf Eisen<sup>1,2</sup>, Holger Schmithüsen<sup>1</sup>, Stefanie Arndt<sup>1,3</sup>, Guido Ascenso<sup>4</sup>, Linda Ort<sup>1,\*</sup>, and Zsófia Jurányi<sup>1</sup>

**Correspondence:** Valerie Reppert (valerie.reppert@awi.de)

Abstract. This study investigates monthly snow accumulation derived from in-situ measurements at Neumayer Station, coastal Dronning Maud Land, East Antarctica, over a 33-year period (1991–2024). Snow accumulation is the major component of the surface mass balance, which is among the most uncertain factors of Antarctica's contribution to global sea level rise. The analysis aims to (1) quantify seasonal contributions and detect climatological shifts, (2) compare annual accumulation rates across three measurement sites, and (3) investigate the magnitude and nature of interannual variability. Results reveal high intra- and interannual variability without a consistent seasonal cycle. Out of the four seasons, only the austral autumn season has shown a statistically significant increase in accumulation over the past 30 years. Although no robust long-term trend was detected in annual accumulation rates, the years 2021 and 2023 stand out as statistically rare positive extremes observed across the measurement sites. Spectral analyses reveal pronounced interannual to decadal variability, which hinders the detection of potential trends and raises the question of whether these extremes reflect constructive interference of natural variability modes or indicate the onset of a regime shift in accumulation driven by global climate warming. Supplementary analysis of monthly average meteorological parameters (temperature, relative humidity, and wind fields) revealed no consistent link to accumulation on monthly scales, suggesting a decoupling between local meteorology and snow accumulation at these time scales. This highlights the need for further research into short-term processes and event-scale accumulation drivers. The datasets presented here provide a long-term base for validating regional climate models and for ground-truthing remote sensing products related to Antarctic snow accumulation and surface mass balance.

#### 1 Introduction

The changing climate and the resulting sea level change (SLC) have become critical concerns in both environmental science and policy. As communities and economies are already facing severe consequences, accurate projections of future SLC are essential for informing effective mitigation and adaptation strategies with regard to coastal resilience. Despite significant ad-

<sup>&</sup>lt;sup>1</sup>Alfred Wegener Institute Helmholtz Center for Polar and Marine Research, Bremerhaven, Germany

<sup>&</sup>lt;sup>2</sup>Faculty of Geosciences, University of Bremen, Bremen, Germany

<sup>&</sup>lt;sup>3</sup>University of Hamburg, Institute of Oceanography, Hamburg, Germany

<sup>&</sup>lt;sup>4</sup>CMCC Foundation—Euro-Mediterranean Center on Climate Change, Italy

<sup>\*</sup>Now at: Max Planck Institute for Chemistry, Atmospheric Chemistry Department, Mainz, Germany

vances in climate modeling, substantial uncertainties in process-based physical projections persist, most notably regarding the contribution of the Antarctic Ice Sheet (AIS) mass balance to global SLC (IPCC, 2021, Chapter 9, Table 9.8).

Mass balance is determined by the interplay between surface mass balance (SMB), primarily driven by snowfall and subsequent accumulation, and ice discharge into the ocean. However, even with current technology, SMB cannot be reliably estimated from remote observations alone. Compensating for the limited availability of in-situ observations, regional climate models (RCMs) are therefore widely employed to estimate SMB and its temporal evolution (Ekaykin et al., 2023). These models, however, suffer from systematic biases (Dunmire et al., 2022; Noël et al., 2018; Ekaykin et al., 2023) and tend to underestimate SMB variability (Wauthy and Dalaiden, 2025). For the East Antarctic Ice Sheet in particular, uncertainties in SMB primarily stem from uncertain snowfall rates, with future changes in snowfall remaining ambiguous due to the complex interplay of various climatic drivers (Shepherd et al., 2018).

Accurately estimating present and future SMB also requires accounting for the inherently heterogeneous spatial distribution of snow accumulation influenced by multiple processes that operate across a range of spatial scales. Accumulation becomes typically more heterogeneous over complex terrain, but even on nearly flat surfaces such as ice shelves, snow is rarely distributed uniformly. This variability starts with precipitation, which is generally more evenly distributed than the resulting snow depth (Eisen et al., 2008). During and after snowfall, snow distribution is further modified by wind redistribution, sublimation, melting, and compaction (Clark et al., 2011; Voordendag et al., 2024; Lenaerts et al., 2010; Schlosser et al., 2002). The large-scale spatial pattern of snowfall is primarily governed by atmospheric circulation and topography, leading to variability at scales of several kilometers or more (Clark et al., 2011). However, it is not only the snowfall itself but also the subsequent deposition that is strongly influenced by the interaction between the local wind field and topography (Frezzotti et al., 2005). In addition, obstacles to the wind field, such as anthropogenic constructions or icebergs in near-coastal sea ice, can significantly modify local accumulation patterns by creating wind shadows that affect snow deposition both upwind, downwind, and laterally (Hames et al., 2025; Franke et al., 2025; Stefanini et al., 2025). Spatial contrasts are typically more pronounced where slopes vary along the prevailing wind direction, while flatter terrain or areas of high accumulation tend to exhibit lower variability (Frezzotti et al., 2005). At the meter scale, sastrugi, wind-formed ridges of hardened snow, introduce substantial local heterogeneity (Eisen et al., 2008). However, because single sastrugi are confined to small areas (on the order of 10 m), their influence is commonly reduced through spatial averaging (McConnell et al., 1997). Over longer distances, spatial variations in accumulation can equal or even exceed temporal fluctuations observed on multi-decadal to centennial timescales (Eisen et al., 2008; Frezzotti et al., 2005).

Temporal changes in snow depth, measured as the change in snow height, and snow density represent the cumulative effect of the aforementioned processes (van Den Broeke et al., 2004), i.e. the net result of accumulation (mass gains), ablation (mass losses), and compaction (no net mass change). In the following, we use the term accumulation to denote the observed accumulated snow height change, expressed in dimension of length. This deviates from the strict mass-based definition of accumulation, but is justified in the context of surface observations, which record height changes rather than mass directly. Negative accumulation therefore reflects the combined effects of ablation and compaction. At a given location, the specific surface mass balance rate  $\dot{b}$  can be expressed as the product of the mean snow density  $\bar{\rho}$  and the accumulation rate  $\dot{a} = \frac{\Delta a}{\Delta t}$ ,

60

where  $\Delta a$  is the accumulation over a given time interval  $\Delta t$ :

$$\dot{b} = \overline{\rho} \frac{\Delta a}{\Delta t}.\tag{1}$$

By integrating this specific rate  $\dot{b}$  over a defined area A, such as that of a glacier or ice sheet, we obtain the total SMB, often denoted as  $\dot{B}$  (Cuffey and Paterson, 2010).

Temporal variability of snow accumulation is influenced by three primary mechanisms: thermodynamic processes, synoptic-scale dynamics, and large-scale modes such as El Niño Southern Oscillation (ENSO) and Southern Annular Mode (SAM) (Dalaiden et al., 2020). Some studies suggest large-scale modes, especially SAM, dominate regional mass balance and SMB variability (King et al., 2023; Medley and Thomas, 2019). Others find strong year-to-year variability, for instance in the ENSO signal in SMB (Macha et al., 2024; Ayabilah et al., 2025), highlighting uncertainties in the relative contributions of different processes. On annual timescales, Turner et al. (2019) found that up to 70% of the variance in annual precipitation can be attributed to extreme precipitation events (EPEs), with a stronger influence observed over ice shelves compared to the plateau. These EPEs are typically associated with synoptic-scale intrusions of maritime air masses or even atmospheric rivers (Gorodetskaya et al., 2014, 2020). Information on intra-annual accumulation patterns remains particularly limited due to the scarcity of high-temporal-resolution datasets; studies conducted in Dronning Maud Land (DML) show weak seasonality in accumulation driven mainly by four to five snowfall events per year (Reijmer and van den Broeke, 2003).

Here we analyze a continuous 33-year record of in-situ snow accumulation measurements in the vicinity of Neumayer Station, East Antarctica's DML, combining data from stake farms, a laser distance sensor, and an ultrasonic snow buoy. This rare long-term and high-resolution dataset allows us to quantify seasonal and interannual variability in snow accumulation, assess the spatial representativeness of point measurements, and explore potential links between accumulation patterns and large-scale climatic drivers. By analyzing trends, extremes, and spectral properties of the accumulation time series, we aim to distinguish natural variability from anthropogenic influences and assess local meteorological effects. These findings provide critical ground-truth data to validate climate models and remote sensing, refining Antarctic mass balance projections. This paper is organized as follows: Section 2 describes the measurements and applied statistical methods; Section 3 presents the spatial and temporal analysis of accumulation rates; Section 4 discusses the results in a broader climate context; and Section 5 provides conclusions.

# 2 Methods

#### 2.1 Instrumental Data

Snow accumulation on the Ekström Ice Shelf, DML, Antarctica, has been measured using three approaches: stake farms, a laser distance sensor, and an ultrasonic distance sensor (Table 1). Since 1981, three successive Neumayer research stations have been located on the ice shelf: Georg von Neumayer (1981–1992), Neumayer II (1992–2009), and Neumayer III (since 2009, NMIII, Fig. 1, (Alfred-Wegener-Institut Helmholtz-Zentrum für Polar- und Meeresforschung, 2016)). In the following we present technical details, basic processing and discuss limitations of each method.

**Table 1.** Overview of the available snow accumulation datasets used in this study. The stake farms consist of multiple individual stakes arranged in regular grids. Abbreviations: Stake farms are named Süd, Neumayer (NM), and Spuso. The laser sensor is SHM (Jenoptik SHM30), and the ultrasonic sensor is the Snow Buoy.

|                                      |                     | Stake Farms         |          | <u>Laser</u> | <u>Ultrasonic</u> |
|--------------------------------------|---------------------|---------------------|----------|--------------|-------------------|
|                                      | Süd                 | NM                  | Spuso    | SHM          | Snow Buoy         |
| Period                               | Dec 1990            | Mar 1992            | Sep 2009 | Jan 2013     | Feb 2013          |
| 1 criod                              | Mar 2025            | Jan 2009            | Mar 2025 | Dec 2024     | Mar 2023          |
| Time reading interval                | 2–4 weeks           | 1 week              | 1 week   | 1 minute     | 1 hour            |
| Measurement points                   | 16 (4×4)            | 25 (5×5)            | 16 (4×4) | 1            | 4                 |
| Stake spacing [m]                    | 10                  | 10                  | 10       | -            | 1                 |
| Footprint of single measurement [cm] | 1–10                | 1–10                | 1–10     | 0.1–1        | 50–100            |
| Comment                              | until Sep 1994 only | until Feb 1994 only |          |              |                   |
| Comment                              | farm average        | farm average        |          |              |                   |

#### 2.1.1 Stake Farm Measurements

The Alfred Wegener Institute (AWI) established three stake farms at different locations on the Ekström Ice Shelf. These sites drift towards the open sea at different speeds that are color-coded in Fig. 1(b).

Süd: Located 6 km southwest of NMIII (70°40′S, 8°16′W), drift speed:  $\approx 178 \text{ m yr}^{-1}$ ; total displacement  $\approx 6.4 \text{ km}$  since 1990.

NM: Located at AWI's former research station Neumayer II ( $70^{\circ}39'S$ ,  $8^{\circ}15'W$ ), drift speed:  $\approx 146 \text{ m yr}^{-1}$ ; total displacement  $\approx 4.9 \text{ km}$  since 1992.

95 *Spuso*: Located at the Air Chemistry Observatory ("Spuso") site , 1.5 km south of NMIII, drift speed:  $\approx$  158 m yr<sup>-1</sup>; total displacement  $\approx$  2.6 km since 2009.

Süd is the stake farm with the longest time series, while the stake farm NM was originally located near the former Neumayer Station II and discontinued upon moving the station and was replaced by the third stake farm Spuso, now located 1.5 km south of NMIII.

For stake farm measurements, stakes are inserted into the snow at regular spacing with a known length protruding above the surface. Snow accumulation is tracked by periodically measuring the exposed length of the stake, assuming the stake base remains fixed relative to the surrounding snowpack. The aluminum stakes are 3 m long, with a diameter of 30 mm and a wall thickness of 3 mm.

Multiple stake measurements are used to reduce small-scale depositional variability and improve signal-to-noise ratio. Beyond improving signal quality through averaging, stake farms provide valuable insights into the spatial variability of snow accumulation at small scales and offer a means to assess the spatial representativeness of individual measurements.

**Figure 1.** (a): Inset: Map of Antarctica with a purple rectangle indicating the study area. The red sector marks Dronning Maud Land. Main map: Zoom-in on the Neumayer Station site. Black lines represent isohypses from the RAMP2 (Liu et al., 2015) digital elevation model, and ice flow velocity from the MEaSUREs dataset (Rignot et al., 2017) is shown as color-codeding and vectors. (b): The Stake Farm "Süd" is located on the moving Ekström Ice Shelf; the series of dots indicates the reconstructed, interpolated trajectory of the stake farm over the 33-year measurement period, showing its gradual movement towards Neumayer Station. While the other sites also drift, their trajectories are not shown in the figure. Basemap and datasets from Quantarctica v3 (Matsuoka et al., 2018). (c): Photograph of the stake farm Spuso. Photo: Linda Ort, used with permission.

# 2.1.2 Laser and Ultrasonic Distance Sensors

Distance sensors work in principle as altimeters, emitting a pulse and measuring the return time. With known wave speed in air the distance of the surface reflector can be determined. Such instruments have a limited spatial footprint, representing a small area of a few square meters at most. Laser distance sensors (Jenoptik SHM30/ Lufft SHM31, hereafter SHM, (Schmithüsen, 2023)) were operated with 1-minute sampling intervals, providing high temporal resolution snow accumulation data. The sensors (one at a time) were mounted on a mast 200 m west of the Spuso stake farm.

Approximately 20 m from the SHM, a so-called Snow Buoy (Nicolaus et al., 2021) was deployed. This system comprises four ultrasonic distance sensors (HRXL-MaxSonar-WR3) arranged in a square layout at 1 m distance and sampling at 1-hour intervals. The sensors are mounted on a 1.75 m mast, with measurement heads positioned at 1.5 m above the surface. The buoy is regularly dug out and reset at a higher level on the snow to ensure reliable measurements. To increase the spatial representativeness for the measurement, the median of the four ultrasonic sounders is used as a combined signal of accumulation.

Both, the SHM and Snow Buoy datasets were filtered using a 24-hour moving median filter, following the approach by Steiner et al. (2023) to remove outliers. In addition, extreme outliers associated with blowing or drifting snow events were filtered (SHM: 0.0%; Snow Buoy: 0.31%). A custom threshold was applied to remove implausible rapid accumulation events, defined as changes exceeding 70 cm within 10 hours, which is unlikely in open terrain given the annual accumulation of approximately 1 m per year. A daily accumulation rate was derived from the filtered sub-daily measurements by sampling for each day the last available value to compute daily changes (i.e., surface height change as accumulation increments with respect to the previous day).

# 125 2.1.3 Height Correction

Snow densification above the stake or mast bottom leads to an underestimation of accumulation in snow height change measurements. The corrected accumulation  $\Delta a$  is therefore given by:

$$\Delta a = \Delta H_{fc} + \Delta h,\tag{2}$$

where  $\Delta h$  is the measured snow height change and  $\Delta H_{fc}$  is the lowering of the snow surface due to firn compaction (subscript fc). Following the arguments of Takahashi and Kameda (2007), this correction term can be calculated as (Eisen et al., 2008)

$$\Delta H_{fc} = \dot{b} \left( \frac{1}{\rho_0} - \frac{1}{\rho_b} \right) \Delta t,\tag{3}$$

where  $\Delta t$  is the time interval over which  $\Delta H_{fc}$  is evaluated (one year in this study). All other variables represent annual mean values:  $\overline{\rho_0}$  the surface snow density, and  $\overline{\rho_b}$  the snow density at the depth corresponding to the bottom of the stake or mast.

As density measurements were not consistently performed at the stake farms, especially not  $\overline{\rho_b}$ , we use the empirical densification model by Herron and Langway (1980) to estimate  $\overline{\rho_b}$  from a modeled density–depth profile. The model requires inputs of  $\overline{\rho_0}$ , annual mean accumulation (converted to kg m<sup>-2</sup> a<sup>-1</sup>), and mean annual snow temperature, which we approximate by using the 2 m air temperature. Based on extensive measurements at stake farms NM, Spuso, and Süd we follow Hecht (2022) and take an average surface density of  $\overline{\rho_0} = 396$  kg m<sup>-3</sup>. Accumulation rates are approximately 1 m a<sup>-1</sup>, with a first-meter mean density  $\overline{\rho_{1m}} = 422$  kg m<sup>-3</sup>. This yields the required input for firn densification modeling.

The 3 m long stakes were replaced when the remaining visible part above the surface got too short, thus an average bottom depth of 2 m is assumed throughout the record. This results in a constant correction of  $\Delta H_{fc} = 9.5$  cm for all stake-based annual accumulation values.

In contrast, the SHM mast is extended when necessary, with a base fixed to the surrounding firn layer, leading to a progressively higher value of  $\rho_b$  and thus an increasing correction  $\Delta H_{fc}$  over time. To validate the correction approach, we include data from a nearby snow buoy equipped with four ultrasonic sensors at 1.5 m height. As the buoy is ramped up regularly, we consider it least affected by firn densification. Nevertheless, we apply a fixed  $\rho_b$  corresponding to a depth of 1 m. Sensitivity tests show that varying the input 2 m air temperature in the densification model by  $\pm$  1 K has only minor influence on the correction term (less than 3%).

Applying the correction significantly reduces discrepancies between SHM, stake farm, and buoy-derived cumulative accumulation (Fig. 2). For the SHM mast, the comparison between raw and corrected data indicates that approximately 3 m of



**Figure 2.** Cumulative annual snow accumulation at the Spuso observatory at NMIII in the period 2013–2025. Data are shown for SHM (orange), the Spuso stake farm ("Stakes", yellow), and the nearby snow buoy ("Snow Buoy", cyan). For each dataset, both uncorrected (raw accumulation measurements, dashed lines) and corrected values (adjusted for snow compaction based on an annual mean density profile, solid lines) are presented.

surface lowering occurred due to compaction between 2013 and 2024. While the analysis of interannual variability is robust even without density corrections, reconciling long-term snow mass fluxes and comparing records derived from different measurement techniques requires accounting for snow compaction. Comparing accumulation across stake farms, however, remains valid without height correction, assuming comparable surface densification conditions and similar  $\rho_0$ , as demonstrated by Hecht (2022) for the stake farms Süd, NM and Spuso. The majority of the variability in accumulation reflects precipitation variability, with compaction-related height changes contributing only marginally. This also holds for SMB variability more broadly: at Vostok Station (78°28'S, 106°50'E, 3,488 m a.s.l.), an East Antarctic research station with stake farm measurements, and stake farm Süd, accumulation changes account for approximately 90% of the observed variability, while changes in snow density contribute only about 10% (Ekaykin et al., 2023; Reppert, 2024). Given this, the height correction is applied only to analyses conducted at the annual scale where absolute snow mass comparisons are made. In all other investigations that focus on relative changes or variability, the correction is omitted.





## 2.1.4 Meteorological Data

To support the analysis of snow accumulation, meteorological observations from NMIII were used. Specifically, air temperature (2 m and 10 m), relative humidity, and wind speed and direction (2 m and 10 m) from January 1990 to January 2022 were considered (Schmithüsen, 2023). The measurement interval changed over time from 10 minutes (prior to 1992) to 5 minutes (from 1992), and to 1 minute (from 1998). All parameters were aggregated to monthly averages.

## 2.2 Quality Criteria

We implement a two-stage quality control procedure to assess the reliability of individual accumulation measurements  $\Delta a_{i,j}$  from each stake i in the stake farm of the j-th measurement. If a value is identified as erroneous, it is excluded from the calculation of the mean accumulation  $\Delta a_j$  for the respective stake farm. To detect such outliers, two quality flags are applied. First Quality Flag: Statistical Deviation For each measurement j, we first compute the mean accumulation  $\Delta a_j$  across all stakes i. To mitigate the potential influence of outliers on the standard deviation, we utilize a constant standard deviation  $\tilde{\sigma}$  calculated from the entire time series, rather than computing a measurement-j-specific standard deviation  $\tilde{\sigma}_j$ . This approach assumes that spatial variability in snow accumulation remains relatively consistent over time. A stake measurement is flagged if its accumulation  $\Delta a_{i,j}$  deviates more than 2  $\sigma$  from the mean, i.e., falls outside the interval  $[\Delta a_j \pm 2\tilde{\sigma}]$ .

Second Quality Flag: Extreme Value Detection The first flag has a limitation: adjacent stakes deviating significantly in the same direction may indicate sastrugi rather than measurement error. To address this, we implement a second quality control mechanism focused on single extreme value detection. For each measurement day, we compare the most extreme accumulation value with the second most extreme value and the remaining stakes. Specifically, we calculate:

- the difference between the maximum (or minimum) accumulation  $\Delta a_{i,\text{max}}$  and the second-highest (or second-lowest)  $\Delta a_{i,\text{max}2}$ ;
  - the difference between the second-highest (or second-lowest) and the minimum (or maximum)  $\Delta a_{i,\min}$ .

If  $\Delta a_{i,\text{max}} - \Delta a_{i,\text{max2}}$  exceeds  $\Delta a_{i,\text{max2}} - \Delta a_{i,\text{min}}$ , the extreme value is flagged as potentially erroneous. To be rejected, a stake measurement had to fail both quality criteria. Applying this procedure resulted in the exclusion of 17 out of 10384, 25 out of 12928, and 21 out of 21875 individual measurements from the Süd, Spuso, and NM stake farms, respectively. The field operators manually measured the exposed stake length and subsequently transferred the values into an Excel sheet, with the main source of uncertainty arising from manual data transcription.

#### 2.3 Temporal Adjustments

The datasets span different time periods, with stake farm Süd providing the longest continuous record (since 1990). All stake farm measurements were conducted by overwintering personnel during favorable weather conditions (Franke et al., 2022). At Süd, the temporal measurement interval increased from monthly to bi-weekly readings since 2010. Higher measurement frequency typically yields lower accumulation values per measurement interval. Therefore, all datasets were resampled and

interpolated to a common monthly resolution, which corresponds to the measurement frequency of the Süd stake farm prior to 2010. To correct for irregular intervals, accumulation rates are normalized by the duration of each interval. Here,  $t_j$  denotes the time of the j-th observation and on the irregular time axis, and  $\Delta t_j = t_j - t_{j-1}$  is the number of days between two consecutive measurements. The corresponding accumulation  $\Delta a_j$  is divided by  $\Delta t_j$ , yielding a daily accumulation rate  $\dot{a}_j$  in cm per day:

$$\dot{a}_j = \frac{\Delta a_j}{\Delta t_j}.\tag{4}$$

These daily accumulation rates are subsequently averaged onto a regular monthly axis  $\dot{a}_m$  by weighting each rate according to the number of days it represents:

$$\dot{a}_m = \frac{1}{n_m} \sum_j \dot{a}_j \, n_{m,j},$$
 (5)

where  $n_m$  is the number of days in month m, and  $n_{m,j}$  is the number of days in month m for which the measurement at  $t_j$  and its derived daily accumulation rate  $\dot{a}_j$  is applicable (with  $\sum_j n_{m,j} = n_m$ ). Thus, the resulting  $\dot{a}_m$  also describes accumulation per day in cm day<sup>-1</sup>. While the original weekly data from NM and Spuso frequently recorded 0 cm of accumulation punctuated by occasional extreme values, temporal downsampling to monthly resolution inevitably leads to a loss of detail regarding the precise timing of individual accumulation events. Nevertheless, elevated monthly accumulation rates still reflect the occurrence of extreme events within the respective month. As the primary objective of this study is to construct a robust and comparable accumulation climatology near Neumayer Station, this trade-off is considered acceptable. Moreover, the higher temporal resolution of the original data can still be fully utilized in analyses of detrended cumulative accumulation. Since cumulative values are derived through temporal integration, the precise resolution of the input time series becomes less relevant in this context.

## 210 2.4 Spatial Analysis




To evaluate the temporal and spatial representativeness of the SHM, we used the Spuso stake farm as a reference. We computed the Pearson correlation coefficient r and coefficient of determination  $R^2$  between the time series of each individual stake and the farm-average time series. This analysis was then repeated for the SHM time series and the farm average. This approach allowed us to quantify both the general explanatory power of single-point measurements and the specific performance of the SHM in capturing observed variability. Due to the higher temporal resolution of the SHM (daily) compared to the Spuso stake farm (approximately weekly), we cumulatively summed SHM accumulation values up to each stake farm measurement day. This yielded a temporally aligned time series, from which r and  $R^2$  could be consistently computed. Given the weekly temporal resolution of this analysis, compaction effects are considered minimal and were not height corrected. As the Pearson correlation is invariant to linear biases and compaction acts similarly across the Spuso stake farm and the SHM, the impact on correlation metrics is negligible.

## 2.5 Temporal Analysis

To quantify uncertainties in trends and averages of annual accumulation rates  $a_{\Sigma}$ , we compute 95% confidence intervals (CIs) of the temporal mean  $\overline{a_{\Sigma}}$  at a significance level of 0.05. To quantify interannual variability, we computed the temporal standard





deviation of the annual mean accumulation ( $\overline{a_{\overline{\Omega}}}$ ). Relative variability was expressed as the ratio of  $\sigma_{\overline{a_{\Sigma}}}$  to the long-term mean annual accumulation ( $\overline{a_{\Sigma}}$ ). For annual means, we assume an underlying Student's t-distribution, which is appropriate given that all accumulation time series consist of fewer than 35 years of data. This assumption is supported by the fact that annual accumulation rates do not reject the null hypothesis of normality, as verified by the Shapiro–Wilk test (Wilks, 2011). In contrast, the monthly accumulation rates do reject the normality assumption, and their histograms exhibit deviations from a Gaussian distribution. Therefore, for monthly statistics, we estimate the CIs using a non-parametric bootstrapping approach (Wilks, 2011). The CIs are reported in square brackets [ , ] following the respective values. At the 5% significance level, we consider differences between means to be statistically significant when their 95% CI do not overlap. To estimate linear trends, we apply ordinary least squares regression to seasonal averages of accumulation rates and cumulative accumulation. For the latter, the regression is used to remove the long-term trend, effectively serving as a low-pass filter. Notably, cumulative snow accumulation can be computed without resampling the time series to uniform time intervals. To evaluate the strength of association between variables, such as accumulation measurements from different sites or measurement methods, we primarily use Pearson's correlation coefficient, assuming a linear relationship.

Spectral characteristics of the accumulation time series are examined using both Fast Fourier Transform (FFT) and wavelet analysis, which serve as complementary tools for identifying periodic features. These analyses are performed on both the original and detrended cumulative time series at monthly resolution. For the FFT, we account for the different measurement periods by defining a cut-off frequency, which corresponds to one-fourth of the total time series length, assuming a minimum of four data points is required to accurately resolve a periodic signal. For wavelet analysis, we employ the Python implementation provided by Torrence and Compo (1998), which by default uses a Morlet wavelet, which is a common choice in geophysical applications.

## 2.6 Analysis of SAM and ENSO Effects on Snow Accumulation

To assess potential linear relationships between local snow accumulation and large-scale climate modes, Pearson correlation coefficients were calculated between the detrended monthly cumulative accumulation record at Süd and detrended monthly cumulative indices of SAM and Southern Oscillation Index (SOI), representing the ENSO variability. Finally, to investigate potential causal influences of these large-scale climate modes on local snow accumulation, we calculated the transfer entropy (TE) between the detrended monthly cumulative accumulation record at Süd and selected teleconnection indices—namely the ENSO indices nino12, nino3, nino4, and nino3.4, and the SAM, using the indices in their original, detrended, and cumulative forms. TE, which is an information-theoretic generalization of Granger causality (Granger, 1969), quantifies the degree to which the past states of one variable reduce the uncertainty in predicting the future state of another, beyond what is already explained by the latter's own history. In other words, TE tests whether a predictor X (here, the indices) helps forecast the future values of a predictand Y (here, snow accumulation) beyond what an autoregressive model AR(p) of Y would already predict (Barnett et al., 2009). Unlike linear correlation or regression approaches, TE captures both linear and nonlinear, potentially lagged dependencies, and is therefore well suited to detect linkages between remote climate drivers and local

hydro-meteorological responses. In this context, TE provides a robust, model-free framework to test whether variability in accumulation at Süd could be statistically driven by fluctuations in large-scale teleconnections.

#### 3 Results






In this section we will present results from spatial and temporal analysis of snow accumulation measurements at Neumayer Stations. The spatial analysis examines correlation patterns between stake pairs to assess the representativeness of point measurements and identify potential spatial anisotropy in snow accumulation. The temporal analysis will quantify long-term trends and variability across different timescales, from seasonal cycles to decadal changes.

# 3.1 Spatial Analysis

To assess the small-scale spatial coherence of snow accumulation, we analyze the correlation between individual stake pairs as a function of distance and orientation. Figure 3 shows variograms of snow accumulation for the three stake farms. The analysis is divided into the periods 1995-2008 (comparing Süd and NM) and 2010-2024 (comparing Süd and Spuso). In the first period (1995–2008), correlations between stake pairs at NM and Süd are comparable when monthly snow accumulation is considered. Without resampling NM to the same monthly resolution, correlations at NM would appear lower on average due to the higher temporal variability captured by weekly measurements (see Fig. S2 in the Supplement). The difference in correlation between NM and Süd remains small across all distances. While the variogram curves for both farms are relatively noisy—partly due to the limited number of pairwise combinations influencing the average correlation—there is a clear decay of correlation with increasing distance, indicating similar spatial decorrelation patterns. When disaggregating by stake pair orientation, a pronounced directional dependence becomes evident. Stake pairs aligned E-W show higher correlations (by more than 0.1) than those aligned N-S. This pattern is consistent across both Süd and NM. For diagonal orientations (NW-SE and NE-SW), correlation differences are minimal at short distances but become more pronounced at larger separations. Interestingly, the pattern of diagonal anisotropy differs between the two farms: at Süd, NW-SE pairs are more strongly correlated than NE-SW pairs, while the opposite is observed at NM. In the second period (2010–2024), the variogram curves are smoother overall. Again, both Süd and Spuso show a clear decay of correlation with distance, with Spuso being slightly more correlated than Süd  $(\Delta r 

**Figure 3.** Variograms of correlations in monthly snow accumulation between stake pairs as a function of distance for the stake farms NM, Süd, and Spuso. (a): comparison between Süd and NM for 1995–2008. (b): comparison between Süd and Spuso for 2010–2024. Markers indicate the orientation of stake pairs (e.g., — :E–W or / : NE–SW), and shaded areas represent the range between individual orientation correlation coefficients. Numbers next to data points indicate the number of stake pair combinations (n) contributing to each mean correlation coefficient. Abbreviations: N–north, S–south, W–west, E–east

located 200 m west of the stake farm, this finding is consistent with the spatial patterns revealed by the variograms. Notably, two stakes within the Spuso farm separated by only 42m share as little as 12% common variability on weekly time scales (see Fig. S2 in Supplement), highlighting the strong spatial heterogeneity of accumulation within the farm. These results indicate that the SHM captures accumulation characteristics at Spuso only to a limited extent on weekly or shorter time scales. However, its high temporal resolution provides valuable complementary information for investigating accumulation processes that cannot be resolved by weekly stake measurements alone.

#### 3.2 Temporal Analysis



# 3.2.1 Monthly and Seasonal Accumulation Rates

To examine potential seasonal patterns and long-term changes in accumulation behavior, we analyzed the monthly accumulation rates across the entire observation period for intra-annual cycles and potential shifts thereof. The monthly accumulation

Figure 4. (a): Heatmap of monthly average accumulation rates (cm/day) for stake farm Süd, 1991–2024, where red (blue) cells represent months of net ablation (accumulation). (b): Annual mean accumulation rates shown as green bars (cm/day) with standard deviations ( $1\sigma$ ) from the same intra-annual values used to compute the means as grey error bars. (c): As for (b) but for monthly mean accumulation rates and their standard deviation reflecting interannual variability.




rates at Süd for each year between 1991 and 2024 (Fig. 4) provide the basis for the following analyses and are representative of the behavior across the presented accumulation time series also at Spuso, NM and SHM. This is supported by correlations of Süd with Spuso (r = 0.73), SHM (r = 0.70), and NM (r = 0.58), indicating that they share the same variability signal, with differences mainly in magnitude rather than timing. Monthly accumulation rates exhibit the following characteristics:

- a) accumulation is predominantly positive,
- b) variability is high on both interannual and intra-annual timescales, and
- c) no consistent seasonal cycle is apparent—although ablation most frequently occurs during the austral summer months (December–January), these months can also experience some of the highest accumulation rates (e.g., January 2005 was the third-highest monthly accumulation in the entire time series).

The absence of a robust seasonal cycle leads to substantial interannual variability in the monthly accumulation rates (Fig. 4(c)). In fact, the standard deviation often matches or exceeds the mean for both the monthly and annual averages (Fig. 4(b)). However, grouping monthly mean accumulation rates by season, a more consistent pattern emerges: autumn (MAM) exhibits the highest accumulation rates, followed by spring (SON) and winter (JJA) (see Table 2 and Fig.4(c)). Summer (DJF) consistently shows the lowest accumulation across all stations, except at NM, where summer was the second-highest accumulation season during the 1993-2008 period. When the time series at Süd is subset for this period, elevated summer accumulation becomes visible as well, although it subsequently decreased (see Table 2). All seasonal averages and their CI overlap, indicating no statistically significant differences in accumulation rates across the sites. The only exception is winter accumulation 2014–2024, where measurements from the stake farm Spuso show significantly lower values compared to Süd. Interestingly, the SHM located adjacent to the stake farm Spuso records winter accumulation rates that agree with those at Süd. The temporal evolution of seasonal and annual accumulation rates at Süd is shown in Fig. 5. These results support the previous observation: summer accumulation temporarily decreased from 2005 to 2015, after which it increased again. However, this trend is not statistically significant. In contrast, autumn accumulation exhibits a more persistent and significant increase, which drives the overall upward trend in annual accumulation rates. This trend becomes statistically significant at  $\alpha = 0.05$  (two-sided Pearson and Mann-Kendall tests) only when including the most recent four years of data. Prior to 2020, the trends remained statistically insignificant, with p-values of 0.06 (autumn) and 0.20 (annual average). As typical for linear regressions over time, the first and last data points exert the greatest leverage, thus the last few years inherently contribute strongly to the significance of the trend, especially since at least two of the last four years show above-average accumulation in all seasons. To assess whether this constitutes a sustained regime shift or merely recent positive anomalies, we also evaluate annual accumulation rates in a broader temporal context.

## 3.2.2 Statistics of annual accumulation rates

Despite the statistically significant increasing trend in annual accumulation rates shown in Fig. 6, the linear trend at Süd would become insignificant ( $\alpha = 0.05$ ) if the 2025 annual accumulation falls below 100 cm. This threshold matches the lower bound

**Table 2.** Seasonally and annually averaged accumulation rates given as cm/day and the corresponding 95% confidence intervals for summer (DJF), autumn (MAM), winter (JJA), spring (SON), and annually.

|       | Period    | DJF           | MAM          | JJA          | SON          | Annual       |
|-------|-----------|---------------|--------------|--------------|--------------|--------------|
| Süd   | 1991-2024 | 0.16          | 0.40         | 0.26         | 0.29         | 0.28         |
|       |           | [0.11, 0.21]  | [0.35, 0.45] | [0.22, 0.31] | [0.24, 0.34] | [0.25, 0.31] |
| Süd   | 2010-2024 | 0.12          | 0.39         | 0.30         | 0.34         | 0.31         |
|       |           | [0.05, 0.19]  | [0.39, 0.55] | [0.24, 0.36] | [0.29, 0.40] | [0.26, 0.35] |
| Spuso | 2010-2024 | 0.11          | 0.46         | 0.22         | 0.35         | 0.29         |
|       |           | [0.05, 0.18]  | [0.37, 0.55] | [0.17, 0.28] | [0.29, 0.42] | [0.25, 0.32] |
| Süd   | 1993–2008 | 0.21          | 0.33         | 0.24         | 0.24         | 0.26         |
|       |           | [0.14, 0.28]  | [0.27, 0.39] | [0.18, 0.31] | [0.16, 0.33] | [0.21, 0.30] |
| NM    | 1993-2008 | 0.20          | 0.23         | 0.11         | 0.15         | 0.18         |
|       |           | [0.15, 0.26]  | [0.16, 0.30] | [0.06, 0.18] | [0.09, 0.22] | [0.14, 0.21] |
| Süd   | 2014–2024 | 0.13          | 0.45         | 0.32         | 0.33         | 0.31         |
|       |           | [0.02, 0.22]  | [0.36, 0.55] | [0.25, 0.40] | [0.26, 0.40] | [0.25, 0.37] |
| Spuso | 2014-2024 | 0.12          | 0.45         | 0.22         | 0.34         | 0.28         |
|       |           | [0.02, 0.21]  | [0.34, 0.56] | [0.15, 0.28] | [0.26, 0.43] | [0.24, 0.33] |
| SHM   | 2014-2024 | 0.02          | 0.37         | 0.30         | 0.23         | 0.23         |
|       |           | [-0.08, 0.14] | [0.28, 0.47] | [0.22, 0.39] | [0.11, 0.36] | [0.19, 0.28] |

Table 3. Height-corrected mean annual accumulation  $\overline{a_{\Sigma}}$  and interannual variability  $\sigma_{\overline{a_{\Sigma}}}$  in cm/a for each site, calculated for different time periods. Confidence intervals are provided at the 95% confidence level. Intervals for the sample mean are based on the student-t-distribution; those for the sample standard deviation are derived from the  $\chi^2$  distribution.

|       | Period    | $\overline{a_\Sigma}$ | CI             | $\sigma_{\overline{a_{\Sigma}}}$ | CI           | Relative Variability |
|-------|-----------|-----------------------|----------------|----------------------------------|--------------|----------------------|
|       |           | (cm/a)                | (cm/a)         | (cm/a)                           | (cm/a)       |                      |
| Süd   | 1990-2024 | 111.0                 | [99.5; 122.6]  | 33.1                             | [26.7; 43.6] | 0.30                 |
| Süd   | 2010-2024 | 121.3                 | [103.7; 139.0] | 32.0                             | [23.4; 50.4] | 0.26                 |
| Spuso | 2010-2024 | 114.3                 | [100.2; 128.4] | 25.4                             | [20.5; 33.5] | 0.22                 |
| Süd   | 1993-2008 | 103.4                 | [85.6; 121.3]  | 33.5                             | [24.8; 51.9] | 0.32                 |
| NM    | 1993-2008 | 73.7                  | [61.0; 86.4]   | 23.8                             | [17.6; 36.9] | 0.32                 |
| Süd   | 2014-2024 | 122.1                 | [98.0; 146.1]  | 35.9                             | [25.1; 62.9] | 0.29                 |
| Spuso | 2014-2024 | 112.9                 | [93.7; 132.0]  | 28.5                             | [19.9; 50.1] | 0.25                 |
| SHM   | 2014-2024 | 113.1                 | [89.6; 136.5]  | 32.6                             | [22.8; 57.2] | 0.31                 |

Figure 5. Time series of annually (a) and seasonally averaged accumulation rates (b–e) at Süd (1991–2024) shown in blue, with the other stations plotted with increased transparency for comparison. Panels represent (b) summer (DJF), (c) autumn (MAM), (d) winter (JJA), and (e) spring (SON). If a significant linear trend (p 

**Figure 6.** (a): Time series of the height-corrected annual accumulation in cm/a of all available accumulation measurements at stake farm Süd (blue), stake farm NM (bars hatched with "xxx"), stake farm Spuso (hatched with "///") and the SHM (hatched with "..."). (b): Histogram of annual accumulation in cm/a for the comparison period 1993–2008 (n=16) for stake farms NM (green), and Süd (blue). The respective mean annual accumulation is displayed as dashed vertical lines; shaded areas represent the respective 95% confidence intervals. (c): As for (b), but for the comparison period 2014–2024 (n=11) for Süd (blue), Spuso (yellow), and SHM (orange and hatched with "...").

of the 95% CI for the long-term mean (1990–2024:  $\overline{a_{\Sigma}}$  = 111.0 cm/a, 95% CI [99.5, 122.6] cm/a). Given the high interannual variability (temporal standard deviation  $\sigma_{\overline{a_{\Sigma}}}$  33.1 cm/a, or a relative variability 30% of the average annual accumulation  $\overline{a_{\Sigma}}$ ), such a value is well within the expected range under a stationary climate.

Therefore, rather than evaluating a long-term trend across the entire period, it is more meaningful to assess the extremity of individual years. One-sided z-tests were performed to determine whether annual accumulation in a given year significantly exceeds the mean of a defined reference period:

- At Süd, the years 2021 and 2023 exhibit significantly higher accumulation than the 1991–2020 climatology (p = 0.004 and p = 0.011, respectively).
- At Spuso, the same years are also significantly above its respective 2010–2024 mean (p = 0.02 for 2021 and p = 0.005 for 2023).
  - For the SHM measurements, 2021 and 2023 likewise exceed the 2014–2024 mean significantly (p = 0.002 and p = 0.021).
  - In contrast, no individual year stands out significantly in the period 1993–2008 at the stake farm NM.

## 3.2.3 Time series of annual accumulation rates

In the following, we compare the height-corrected annual accumulation for the different sites based on values presented in Fig. 6 and Table 3. In the first comparison period (1993–2008), Süd consistently exhibited higher annual accumulation than NM, with the sole exception of the year 2004. On average, the annual accumulation at Süd exceeded that of NM by 29.7 cm/a (Süd: 103.4 cm/a; NM: 73.7 cm/a; Fig. 6(b)). The respective 95% CIs overlap marginally, indicating a statistically insignificant difference in accumulation levels between the two sites. Interannual variability was higher at Süd, with a standard deviation of 35.5 cm/a compared to 23.8 cm/a at NM. However, relative variability was same for both sites.

In the second comparison period (2014–2024), Süd again recorded the highest mean annual accumulation (122.1 cm/a), followed by SHM (113.1 cm/a) and Spuso (112.9 cm/a), with no statistically significant differences among them (Fig. 6(c)). The time series at all three sites vary in a similar pattern. Interannual variability is comparable across sites, with standard deviations of 35.9 cm/a at Süd, 28.5 cm/a at Spuso, and 32.6 cm/a at SHM. Annual accumulation rates at all sites are consistent with a normal distribution according to the Shapiro–Wilk test, despite some deviations visible in the histograms (Fig. 6(b,c)).

#### 3.2.4 Interannual Variability



To assess interannual to decadal variability, we investigate the cumulative snow accumulation and its residuals (i.e., cumulative accumulation minus the regression line), obtained by removing the linear trend (Fig. 7). The close fit ( $R^2 > 0.99$ ) indicates a stable accumulation regime throughout the observational period. Interannual variability becomes evident only after detrending. Interestingly, all time series vary in phase; only the amplitudes of their deviations from the respective long-term means differ. The amplitude at Süd is the highest, but this may simply result from the fact that the linear regression lines for Spuso, NM, and





SHM are derived from shorter observation periods. As a result, they may more closely approximate the observed cumulative accumulation purely due to the shorter time frame.

The variable of interest is the slope of the detrended time series. Periods with a positive (negative) slope indicate above-average (below-average) accumulation, with larger magnitudes reflecting stronger deviations from the long-term mean accumulation rate. Periods with a slope near zero correspond to average accumulation conditions. The first period with above-average accumulation is from 1995–2000, as can be seen at both NM and Süd, followed by below-average accumulation until 2005 and average accumulation until 2009. In the period 2000–2009 especially in the temporally higher (weekly) resolved NM time series we observe sharp peaks, with abrupt increases followed by gradual declines. This indicates the dominant role of short but high intense accumulation phases of one week or shorter in otherwise accumulation-poor phases. The lower temporally resolved accumulation measurements from Süd follow these patterns but cannot fully capture the accentuated increases. In 2009, measurements at NM ended and began at the Spuso stake farm, coinciding with an above-average accumulation phase (2009–2013), after which SHM also started observations. All three datasets capture an extreme accumulation event in May 2013. Unlike previous events, it was not followed by a gradual decline but rather by several months of average accumulation, culminating in a strong ablation event in early 2014. Subsequently, all datasets exhibit a below-average accumulation phase lasting until 2017, which is followed by a sustained above-average phase that continues to the present. Overall, the residuals reveal decadal-scale variability, with consistent transitions between above- and below-average accumulation phases across all sites.

## 3.2.5 Spectral properties of accumulation and meteorological variables

The time series of residuals indicate periodicities in snow accumulation (Fig. 7 (b)). To identify dominant frequencies, we apply both FFT and wavelet analysis. The original time series of accumulation (Süd and SHM) and meteorological measurements are largely characterized by high-frequency signals consistent with white noise, with the exception of pronounced peaks at 12 and 6 months, reflecting annual and semi-annual cycles (Table 4, representative example for wind speed shown in the Supplement Fig. S3). At Spuso, the semi-annual cycle is the most prominent in the original time series, suggesting a biannual driver of accumulation variability. In contrast, the NM original time series shows relatively weak (sub-)annual signals and instead exhibits a dominant period at 41 months, indicating stronger multi-annual variability (Table 4). After low-pass filtering all sites except SHM reveal a consistent periodicity between 30 and 40 months. Specifically, NM retains the 41-month peak in both filtered and unfiltered series, while Spuso and Süd (post-2009) show dominant periods near 31 months. SHM deviates from this pattern, with a persistent 12-month cycle even after filtering (Table 4).

For meteorological variables, multi-annual periodicities are less pronounced (Table 4). Only the northerly and northeasterly wind components exhibit peaks at 20 and 77 months. Most other wind sectors, such as southwest and south, show no dominant low-frequency variability and are best described as white noise. These findings suggest no direct link between the dominant periodicities in atmospheric variables and those in accumulation.

To further examine temporal changes in frequency content, we perform a wavelet analysis on the Süd residual time series, which provides the longest continuous record (Fig. 8). The wavelet spectrum displays variance as a function of time and

**Figure 7.** (a): Cumulative accumulation in cm at stake farms NM (1992–2009, weekly resolution, in green), Süd (1990–2025, 2-4 weekly resolution, in blue), Spuso (2009–2025, weekly resolution, in yellow) and the laser distance meter SHM (2013–2024, daily resolution, in orange) and their respective linear regression lines in gray. (b): As for (a) but detrended cumulative accumulation (cumulative accumulation minus gray regression line).


period, with brighter colors indicating stronger signals and contours denoting the 95% significance threshold. Between 1995 and 2005, the spectrum reveals a significant periodicity at 35–40 months. This signal weakens between 2008 and 2016 but reemerges after 2016 at slightly shorter periods (30–35 months), although below the 95% significance threshold. These results are consistent with the dominant frequencies identified by the FFT analysis, while additionally showing that dominant periodicities are not stationary and can temporarily disappear.

**Table 4.** Periods (in months) corresponding to the maximum amplitudes obtained from FFT of the original and low-pass filtered monthly averaged time series. Accumulation was detrended by removing the linear trend of the cumulative series; for meteorological variables, seasonal components were removed. The cutoff frequency corresponds to one-fourth of the total time series length (assuming at least four data points are needed to resolve a periodic signal). Abbreviations of meteorological variables: wind directions (E = East, N = North, NE = Northeast, NW = Northwest, S = South, SE = Southeast, SW = Southwest, SE = Southeast, SW = Southwest, SE = Southeast, SE = Southeast, SE = Southeast, SE = Southeast, SE = Southwest, SE = Southeast, SE = Southwest, SE = South

| Variable                 | Original  |            | Low-pass | Cutoff |  |  |  |
|--------------------------|-----------|------------|----------|--------|--|--|--|
|                          | First Max | Second Max |          |        |  |  |  |
| Snow accumulation        |           |            |          |        |  |  |  |
| NM                       | 41        | _          | 41       | 41     |  |  |  |
| SHM                      | 12        | 6          | 12       | 36     |  |  |  |
| Spuso                    | 6         | _          | 31       | 46     |  |  |  |
| Süd (all)                | 12        | 6          | 40       | 103    |  |  |  |
| Süd (≥ 2010)             | 12        | 6          | 31       | 46     |  |  |  |
| Meteorological variables |           |            |          |        |  |  |  |
| E                        | 6         | _          | _        | 96     |  |  |  |
| N                        | 12        | _          | _        | 96     |  |  |  |
| NE                       | 12        | 6          | 77       | 96     |  |  |  |
| NW                       | 12        | 6          | 10       | 96     |  |  |  |
| S                        | 12        | _          | _        | 96     |  |  |  |
| SE                       | 12        | _          | 7        | 96     |  |  |  |
| sw                       | _         | _          | _        | 96     |  |  |  |
| W                        | 12        | _          | _        | 96     |  |  |  |
| FF                       | 12        | 6          | 19       | 96     |  |  |  |
| RH                       | 12        | _          | 12       | 96     |  |  |  |
| Т                        | 12        | 6          | _        | 96     |  |  |  |

Figure 8. (a): Time series of residuals from linear trend removal in cm (Süd, 1991—2024). (b): Wavelet Analysis of with the power signal color coded in units of  $\sigma^2$  (cm<sup>2</sup>). Dashed blue contour lines enclose areas, where the power surpasses a background red noise Fourier spectrum at a significance niveau of 95%. The grey hashed area depicts the cone of influence, where edge effects may distort the result.

#### 4 Discussion



## 4.1 Spatial Analysis

To evaluate the spatial heterogeneity and representativeness of accumulation measurements within the stake farms, we analyzed the directional dependence of correlation patterns using variograms. The variograms indicate that stake pairs aligned E–W are systematically more strongly correlated in terms of snow accumulation than those aligned N–S. This anisotropy is consistent with surface features and dominant wind regimes observed in the region. Visual inspections during the data quality checks frequently identified sastrugi aligned along the E–W axis. This orientation is consistent with the two prevailing wind regime modes: easterly moving synoptic-scale cyclones in the circumpolar trough generate strong easterly winds, and supergeostrophic westerly flows arise from a high-pressure ridge north of Neumayer (König-Langlo et al., 1998; König-Langlo and Loose, 2007). Sastrugi tend to form parallel to the dominant and/or strongest wind direction (Fujita et al., 2011), suggesting that wind

Sastrugi tend to form parallel to the dominant and/or strongest wind direction (Fujita et al., 2011), suggesting that wind direction not only shapes surface roughness but also affects the spatial coherence of snow accumulation. As such, higher correlation along the E–W axis reflects wind-aligned redistribution patterns, with snow being transported and deposited along this direction (Arndt et al., 2020).

Any small remaining differences in overall correlation between Süd and NM or Spuso are minor and may reflect local surface or atmospheric heterogeneity rather than sampling artefacts. Süd experiences colder conditions and potentially stronger snow drift, which could slightly enhance spatial decorrelation. Temperature contrasts during katabatic wind events may further modulate local accumulation variability. From a practical perspective, higher inter-stake correlation implies that a missing or faulty measurement at Süd is more likely to be effectively represented by one of the neighboring stakes. In contrast, isolated stake loss at NM or Spuso could result in the loss of more unique, site-specific signals. However, as noted by Ekaykin et al. (2023), overly strong correlation among stakes may reduce the effective signal-to-noise ratio, as similar noise patterns are recorded across all stakes rather than spatially distinct signals. This concern is supported by the fact that the SHM and Spuso stake farms share only 16% of common variability, indicating limited spatial coherence despite a distance of only 200 m.

# 4.2 Temporal Analysis

## 4.2.1 Monthly and Seasonal Accumulation Rates

On the monthly scale, accumulation values do not exhibit statistically significant autocorrelation across time lags (see Fig. S1 in Supplement), indicating that high accumulation in one month does not reliably predict accumulation in subsequent months, seasons, or even the same month in following years. This temporal independence justifies the aggregation of data into seasonal means for trend analysis. Accumulation trends across the observational period are broadly synchronous among the different stations, suggesting the influence of common large-scale drivers and supporting the spatial coherence of accumulation patterns on monthly and longer timescales. Correlation coefficients between stations (r = 0.54-0.71) further substantiate this interpretation. However, sub-monthly comparisons, such as between the SHM and the stake farm, reveal limited spatial coherence (r = 0.4), indicating that short-term accumulation variability is dominated by local effects.








Despite pronounced changes in seasonal accumulation patterns, these trends do not correspond clearly to observed seasonal temperature changes. For example, at Neumayer Station winter temperatures have significantly decreased (see Fig. S4 in Supplement), yet no consistent winter accumulation trend is evident, whereas the autumn accumulation trend has no counterpart in temperature. This decoupling between temperature and accumulation underscores the complexity of the controlling processes. In this context, rather than temperature-controlled SMB, EPEs may play a key role. Turner et al. (2019) and Simon et al. (2024) found that accumulation in coastal DML is primarily driven by EPEs, which tend to peak in autumn. These events could account for the high inter- and intra-annual variability observed in the accumulation record. Simon et al. (2024) further noted that both the frequency and total precipitation from EPEs have increased over the past 40 years. However, it remains uncertain whether this increase is particularly pronounced in autumn, which would help explain the observed seasonal accumulation trend.

Finally, snow compaction processes may further influence the interpretation of seasonal height changes. Hecht (2022) demonstrated that summer snow densities are significantly higher than winter densities. This indicates, that summer snow undergoes stronger compaction, meaning that summer height changes may appear smaller than they truly are in terms of mass. In addition to melt and sublimation—both primarily summer processes (Jakobs et al., 2019)—this enhanced compaction may contribute to the apparent reduction in summer accumulation.

## 4.2.2 Annual Accumulation Rates and Meteorological Drivers

The analysis of annual accumulation rates revealed statistically rare maxima at all three sites in 2021 and 2023. These years also resulted in a weak non-robust positive trend in the time series. Figure 4 shows that the exceptional accumulation in 2021 did not result from isolated extreme events, but rather from a consistent pattern of above-average monthly accumulation. 11 out of 12 months exhibited slightly elevated accumulation rates, while summer ablation was notably weak. A similar pattern is observed in 2023; however, this year also featured more pronounced extremes: At Süd, April 2023 recorded the second-highest monthly accumulation in the whole record, while at Spuso, the two highest values of the entire time series occurred in April and October 2023. These findings indicate that both persistent conditions and individual events contributed to the annual extremes yet with varying degrees of influence. Notably, no corresponding annual changes in temperature or humidity were detected (see Fig. S4 and S7 in Supplement). Again, this highlights the decoupling between local meteorological trends and accumulation variability, not only on the monthly but also on the annual time scale. However, there is evidence that this decoupling may not apply universally across Antarctica. Ekaykin et al. (2023) identified local air temperature as the primary control on longterm SMB, likely due to the stronger Clausius-Clapeyron-driven sensitivity under cold and dry conditions. Similarly, Medley et al. (2018) showed that increased snow accumulation at Kohnen Station is linked to rising temperatures. This suggests that accumulation at Antarctic plateau sites may be more tightly coupled to temperature than at coastal locations. This difference likely reflects contrasting precipitation regimes: at plateau sites, clear-sky precipitation occurs frequently, which requires the local air masses to reach saturation and is therefore inherently temperature-dependent. In contrast, accumulation at coastal stations such as Neumayer is primarily controlled by the frequency and intensity of maritime air intrusions. While temperature





still governs the moisture-holding capacity of these air masses, the dominant driver of accumulation is the synoptic-scale transport rather than the local temperature itself.

#### 4.2.3 Local Meteorological Changes

Over recent decades, wind regimes at Neumayer in winter have shifted markedly. The frequency of easterly winds decreased by about 12%, while westerly and southwesterly winds became more common (see Fig. S6 in Supplement). In parallel, mean winter wind speeds declined by more than 2 m s<sup>-1</sup> at both 2 m and 10 m heights, corresponding to a reduction of over 20% relative to the 30-year mean (see Fig. S5 in Supplement).

These changes are consistent with the circulation patterns described by Klöwer et al. (2014), who found that warm events are typically associated with easterlies, while cold events coincide with southerly or southwesterly katabatic flow. The observed winter cooling trend (-1.3 K per decade; Fig. S4 in Supplement) may therefore be linked to the reduced frequency of easterly winds and the increased prevalence of katabatic flow. Most snowfall events are associated with easterlies (Hames et al., 2025); however, snow accumulation in winter has remained unchanged despite the reduced easterly contribution.

On annual scales, the mean 2 m and 10 m wind speeds show a long-term decline of -0.03 m s<sup>-1</sup> per year, amounting to about 1 m s<sup>-1</sup> over the 35-year period (Fig. S5 in Supplement). At the same time, the percentage of southerly winds at 2 m height decreased from about 15% in 1990 to 12% in 2025, while no trend is evident at 10 m (see Fig. S8 in Supplement). Westerlies increased consistently at both levels, roughly doubling in frequency from 5% to 10% (see Fig. S9 in Supplement). This is noteworthy, as Gorodetskaya et al. (2020) demonstrated that the westerly high wind-speed regime is typically associated with air nearly depleted of water vapor.

Taken together, these shifts in local meteorology cannot be directly linked to the partially asynchronous changes in snow accumulation. This indicates that accumulation variability is influenced not only by local conditions but also by processes operating on shorter temporal scales and larger spatial patterns.

## 4.2.4 Interannual Variability and Periodicities in Accumulation

Our analyses revealed strong interannual variability, with coefficients of variation ranging from 22 % to 32 % across all sites. Detrended cumulative accumulation varies in phase at all sites, suggesting a shared underlying control rather than independent local effects. Around the year 2017, all sites entered a phase of above-average accumulation that continues to the present day. In addition to the decadal-scale transitions between above- and below-average accumulation phases, frequency analyses identified three dominant periodicities (Table 4)

- 6 months—likely associated with the Semi-Annual Oscillation (SAO), which describes the seasonal contraction and expansion of the circumpolar Westerlies. This dynamic leads to a reduction in the number of synoptic cyclones making landfall in Antarctica during summer and winter (Simon et al., 2024).
- 12 months—corresponding to the annual cycle, which is prominent in all meteorological variables. However, this signal is less pronounced at the stake farms NM (dominant mode: 40 months) and Spuso (dominant mode: 6 months).



- 30–40 months (≈ 3–4 years)—detected at all sites except SHM. The absence of this signal in SHM may be due to the relatively short length of the time series or the high temporal resolution, which may obscure multi-year periodicities.

Wavelet analysis confirms the intermittent nature of the 30–40 month cycle (Fig. 8). Fischer et al. (2004) reported a similar 3–4 year periodicity in sea salt aerosol deposition on the DML Plateau, which is associated with the Antarctic Circumpolar Wave. However, this association is considered non-robust. To further explore potential drivers of these periodicities, we next examine the influence of the dominant climate modes SAM and ENSO.

#### 4.2.5 SAM and ENSO as Drivers of Accumulation Variability

The correlation analysis reveals a weak but statistically significant negative correlation between monthly accumulation at Süd and the SOI (r = -0.13, p 

in the future (e.g., Wille et al., 2021), which could further enhance accumulation at coastal Antarctic sites. At this point, only retrospective analysis in the future can shed more light on whether these recent extremes mark a shift in the underlying accumulation regime or remain within the bounds of natural variability.

#### 5 Conclusions







Despite pronounced interannual variability over more than three decades of snow accumulation measurements at coastal DML, no robust long-term trend emerges. Instead, the record highlights the dominance of natural variability, with only the unprecedented extremes of 2021 and 2023 standing out. This emphasizes both the need for long-term, high-resolution observations and the challenge of detecting climate-change signals in coastal Antarctic accumulation records.

This study analyzed snow accumulation time series based on in-situ measurements from stake farms and a laser distance sensor in the vicinity of the Neumayer research site, coastal DML, Antarctica. The dataset comprises four accumulation time series covering 10 to 33 years at varying spatial and temporal resolutions. Stake farm data were used to assess the spatial representativeness of point measurements and to determine the decorrelation of snow accumulation with distance. It was shown that a single-point observation, such as from a laser distance sensor, captures only a limited portion of the temporal (and evidently spatial) variability, though the laser was sampled from minutes to weekly resolution to match the temporal scale of the stake farm measurements. The spatial pattern of accumulation appears highly anisotropic, likely due to the dominance of easterly winds that form sastrugi and induce this climatological imprint on the surface.

Despite strong interannual variability, snow accumulation at all three sites has remained remarkably stable over the past three decades, with high inter-site correlations indicating consistent large-scale atmospheric forcing. Dominant periods detected by spectral analysis suggests that snow accumulation may be modulated by large-scale modes of climate variability, such as the SAM, ENSO or the Antarctic Circumpolar Wave, further investigation is needed.

No robust long-term increase in accumulation was detected for Neumayer, in contrast to findings from other regions in DML, including the plateau and the eastern coast (Medley et al., 2018; Philippe et al., 2016). However, the years 2021 and 2023 stand out as statistically rare accumulation extremes observed synchronously at all sites, pointing to a large-scale atmospheric driver. These events raise the question of whether the region is entering a new accumulation regime under a warming climate. Yet, no concurrent warming trend is observed at Neumayer Station. Moreover, monthly or annual accumulation rates are not directly linked to local meteorological parameters such as temperature, humidity, or wind. This supports the hypothesis that snow accumulation at coastal Antarctic sites is to a lesser extent sensitive to local temperature changes than inland sites, being primarily driven by episodic intrusions of maritime air masses. The differing sensitivities of SMB to warming, for instance, a coastal versus inland setting, warrant further investigation.

Long-term, high-resolution in-situ snow accumulation measurements are critical for improving our understanding of Antarctic climate and mass balance, and they provide rare observational constraints for a range of approaches. RCMs can use such measurements to evaluate and refine modeled precipitation and SMB patterns, which are otherwise unconstrained in data-sparse regions. For example, Glaude et al. (2024) demonstrated that the choice of RCM can lead to a twofold difference in projected





SMB contributions to SLC of the Greenland Ice Sheet by 2100. Remote sensing-based products rely on ground-truth data to validate snow accumulation retrievals from satellites, ensuring that observational biases are minimized. Reanalysis datasets can be benchmarked against these in-situ time series to assess their representation of local and regional snow accumulation, which is often poorly captured due to complex (post)-depositional processes. Together, these applications demonstrate how the dataset presented here can contribute to more reliable projections of Antarctic mass balance and its contribution to global sea level rise.

Code availability. The Python code used for processing the snow accumulation data and reproducing the analyses presented in this paper is publicly available at Zenodo: https://doi.org/10.5281/zenodo.17313860.

Data availability. SHM and Stake Farm "Süd" data have been made publicly available at PANGAEA (Schmithüsen, 2023; Eisen et al., 2025). Stake Farm NM and Spuso data will be made available at PANGAEA. The compilation of density measurements is available at (Eisen, 2024). Meteorological parameters are available through PANGAEA (Schmithüsen, 2023). Snow Buoy data are available here: Nicolaus and Hoppmann (2018). SAM indices are publicly available from https://legacy.bas.ac.uk/met/gjma/sam.html. ENSO indices are publicly available from https://psl.noaa.gov/enso/data.html

Author contributions. VR conducted the analysis, interpreted the data, and wrote the majority of the manuscript. OE initiated the project, supervised and contributed to its conception and data interpretation. HS provided AWI's meteorological data from Neumayer and prepared it for analysis. SA coordinates and curated the snow buoy measurements. GA conducted the causality analysis using transfer entropy and provided interpretation of these results. ZJ coordinates and curated the Spuso measurements. LO conducted the overwintering fieldwork and stake measurements in 2021 and conducted a first analysis of the data. All authors contributed to data interpretation and the manuscript.

Competing interests. At least one of the (co-)authors is a member of the editorial board of The Cryosphere.

Acknowledgements. We are grateful to the air chemistry overwinterers for conducting the stake measurements (Franke et al., 2022). We thank Cedric Hecht for compiling the density measurements and Rolf Weller and Hans Oerter for supervising the accumulation measurements over many years. We further acknowledge the Alfred-Wegener-Institut/Neumayer Station for logistical support (Alfred-Wegener-Institut Helmholtz-Zentrum für Polar- und Meeresforschung, 2016). Artificial intelligence (AI) was utilized only for grammatical correction and formulation optimization.

Financial Support: The field work was supported through AWI grant ID AWI\_ANT\_8. VR was supported by AWI's INSPIRES program.

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
