# Peer review of "Tracking In-Situ Snow Accumulation at Neumayer, Coastal Antarctica: Signs of Climatic Changes in the past 30 Years?"

_EGUsphere, 2025_

## Referee Comment (RC1)

**Review of "Tracking In-Situ Snow Accumulation at Neumayer, Coastal Antarctica:
Signs of Climatic Changes in the past 30 Years?"**

This manuscript presents and analyzes an exciting new data set of in-situ snow accumulation near Neumayer Station, Antarctica covering 1991-2024 from stakes farms, a laser ranger, and ultrasonic distance sensors. The study looks at the spatial correlation in accumulation rates of small areas, as well as a temporal trends in accumulation. The primarily conclusion is that snow accumulation has high spatial and temporal variability at this site and that temporal variability in particular is not well-explained by either local atmospheric variables or large-scale synoptic patterns.

This manuscript is very well organized and written, presents careful and appropriate statistical analyses of the data set, and explores a range of potential drivers for the observed trends in the data. I have very few comments. I think the manuscript is suitable for publication with a few minor revisions.

**Major Comments:**

**[1]** I would be interested to see more discussion or analysis of the drivers of apparent increase in autumn accumulation. In particular, the current discussion posits that an increase in EPEs might drive this trend. That seems like a hypothesis which could be explored with the available data. Of course, the time resolution of the Süd measurements isn't good enough for a definitive test of the hypothesis, but if I squint at Figure 4, it seems like there might be some increase in monthly accumulation variability over the MAM period from the beginning to the end of the record. (If there's not enough there for a meaningful analysis, noting that in the text would also be helpful to the reader.)

**Detailed Comments:**

**Line 23** – it may be worth noting the role of melt in SMB or justifying by citations why it is not important as your specific study site.

**Line 28** - https://agupubs.onlinelibrary.wiley.com/doi/full/10.1029/2022GL099330 would be another appropriate citation relating to model underestimates of SMB variability.

**Line 67** - https://agupubs.onlinelibrary.wiley.com/doi/pdf/10.1029%2F2022GL100585 would be another appropriate citation discussing atmospheric rivers, extreme precipitation, and EPEs as drivers of overall precipitation trends.

**Figure 1** – the text is much too small, especially in the full Antarctica map inset. Please try to increase the label font size to the equivalent of at least 10 pt font at the 100% zoom level. Is there a reason that the drift trajectories for SHM, Snow Buoy, and Spuso are not shown?

**Line 135** – can you comment on the sensitivity of the correction to different firn densification schemes, or to the appropriateness of the H&L model give any available density data from the site?

**Line 191** – the comment hat higher measurement frequency leads to lower accumulation values per measurement interval is an interesting one. Is this a well-known issue? Are there perspectives on why this occurs? Perhaps a citation here would be appropriate.

**Line 284** – it would be interesting to see the scatter plots for the individual stake vs farm average somewhere in the supplement.

**Figure 6** – I think that panel (a) would be easier to interpret as line charts like those shown in Figure 5. But maybe I am missing some key aspect of the data that is not well-represented in that format?

**Line 337** – why 1991-2020 as the climatological reference period at Süd? Specifically, why exclude the last four years of data in this average but not at SHM or Spuso?

**Line 364** – the distinction between looking at slope vs. magnitude was not entirely clear to me as first in this description, since both are described as indicating above or below average accumulation. Perhaps you can clarify that slope shows deviations relative to the few preceding years (so a sort of change in local average) vs. magnitude which shows deviations from the climatological/long-term average. It might also be helpful to comment on the time period over which you find it meaningful to interpret the slope of the anomalies. Are you thinking about interannual slopes, slopes over a few years of data, etc?

**Line 414** – there is a bit of an unexpected jump in the discussion here. Maybe this transition can be clearer if you remind the reader of the specific conclusions you are trying to explain, with some reference back to a figure or table.